# Association of Systemic Steroid Treatment and Outcome in Patients Treated with Immune Checkpoint Inhibitors: A Real-World Analysis

**DOI:** 10.3390/molecules26195789

**Published:** 2021-09-24

**Authors:** Agnese Paderi, Elisabetta Gambale, Cristina Botteri, Roberta Giorgione, Daniele Lavacchi, Marco Brugia, Francesca Mazzoni, Elisa Giommoni, Susanna Bormioli, Amedeo Amedei, Serena Pillozzi, Marco Matucci Cerinic, Lorenzo Antonuzzo

**Affiliations:** 1Medical Oncology Unit, Careggi University Hospital, Largo Brambilla 3, 50134 Florence, Italy; paderi.agnese@gmail.com (A.P.); elisabetta.gambale@gmail.com (E.G.); botteri.cristina@gmail.com (C.B.); roberta.giorgione90@gmail.com (R.G.); daniele.lavacchi@yahoo.it (D.L.); brugiam@aou-careggi.toscana.it (M.B.); mazzonifr@aou-careggi.toscana.it (F.M.); giommonie@aou-careggi.toscana.it (E.G.); serena.pillozzi@unifi.it (S.P.); 2Department of Experimental and Clinical Medicine, University of Florence, Largo Brambilla 3, 50134 Florence, Italy; susanna.bormioli@unifi.it (S.B.); amedeo.amedei@unifi.it (A.A.); marco.matuccicerinic@unifi.it (M.M.C.)

**Keywords:** immunotherapy, immune checkpoint inhibitors, steroid, irAEs, non-small cell cancer, renal cell cancer, melanoma, biomarkers

## Abstract

Background: Immune-related adverse events (irAEs) are inflammatory side effects, which can occur during immune-checkpoint(s) inhibitors (ICIs) therapy. Steroids are the first-line agents to manage irAEs because of their immunosuppressive properties. However, it is still debated whether or when steroids can be administered without abrogating the therapeutic efforts of immunotherapy. Methods: We retrospectively evaluated 146 patients with metastatic non-small cell lung cancer (NSCLC), melanoma and renal cell carcinoma (RCC) treated with ICIs. We assessed the progression-free survival (PFS) of patients treated with steroids due to an irAE compared to a no-steroid group. Results: The early treatment with steroid (within the first 30 days from the beginning of immunotherapy) was not related to a shorter PFS (*p* = 0.077). Interestingly, patients who were treated with steroids after 30 days from the start of immunotherapy had significantly longer PFS (*p* = 0.017). In a multivariate analysis, treatment with steroids after 30 days was an independent prognostic factor for PFS (HR: 0.59 [95% CI 0.36–0.97], *p* = 0.037). Conclusions: This retrospective study points out that early systemic steroids administration to manage irAEs might not have a detrimental effect on patient clinical outcome in NSCLC, melanoma and RCC patients.

## 1. Introduction

Therapeutic intervention with monoclonal antibodies (mAb) that target immune-checkpoint(s) inhibitors(ICIs) is a new and rapidly evolving anti-cancer strategy that is providing profound clinical efficacy in a proportion of cancer patients with several tumor histotypes [1,2]. ICIs are molecules of coinhibitory signaling pathways that act to preserve immune tolerance, yet they are often utilized by cancer cells to elude immunosurveillance. Thus, ICIs are designed to strengthen antitumor immune responses by interrupting coinhibitory signaling pathways and to promote the immune-mediated elimination of cancer cells. Due to their mechanism of action, ICIs can induce inflammatory side effects known as irAEs, which are unique and different from those of conventional anticancer therapies [3].

Steroids, due to their immunosuppressive properties, are the first-line agents for managing irAEs that may arise during, or following, treatment with ICIs. Steroids exert their immune-modulatory effects by acting on T cell activation, differentiation, and migration [4], suppressing the IL-2-mediated activation of effector T cells [5] and increasing regulatory T-cells [6]. Steroids can alter patients’ microbiome [7] and promote M2 macrophage polarization [8]. As a result, steroids exhibit an immune-suppressive action and are hereby associated with worse clinical outcomes when used concurrently in patients treated with anti-programmed death-1 (anti-PD-1) or anti-programmed death ligand-1 (anti-PD-L1) [9]. Furthermore, their use represents an exclusion criterion from most of the ICIs clinical trials. More specifically, 10 mg daily of prednisone-equivalent is the usual permitted steroid dose within clinical studies [10], since doses ≥ 10 mg of prednisone daily are associated with an increased risk of infection [11] and are thus considered immunosuppressive. 

However, even if the detrimental effect observed in clinical outcomes from ICIs could have biological plausibility in light of the steroid immunosuppressive activity, the strength and reliability of the relationship have only been extrapolated from retrospective/post hoc analyses [12]. Moreover, a significant association with worse overall survival (OS) has only been confirmed for baseline steroids administered for the palliation of cancer-related indications such as dyspnea, pain or fatigue, and symptomatic brain metastases [9,12,13,14,15]. Nevertheless, the use of steroids even at doses ≥10 mg to manage cancer-unrelated indications, such as autoimmune disease, did not affect ICIs’ efficacy [12,16,17]. More intriguingly, the use of steroids, even at high doses for the treatment of irAEs, was not associated with a worse clinical outcome in patients with melanoma [18] and NSCLC [19]. Thus, the role of steroid administration during treatment with ICIs is still controversial. In light of these observations, we conducted an observational study to evaluate the impact on outcome of steroid use for the treatment of irAEs in metastatic NSCLC, RCC and melanoma patients treated with checkpoint inhibitors.

## 2. Results

### 2.1. Patient Characteristics

Between March 2016 and March 2020, a total of 146 patients with metastatic NSCLC (*n* = 67), melanoma (*n* = 46) and RCC (*n* = 33) were treated with ICIs (either nivolumab, atezolizumab or pembrolizumab, depending on the histology of the tumor) at our Medical Oncology Unit, Careggi University Hospital (Florence, Italy). Table 1 summarizes patients’ clinical features.

The average age of enrolled patients at the start of immunotherapy was 67 years, ranging between 27 to 91 years; 67.1% (*n* = 98) were male, and 32.9% (*n* = 48) were female. The most frequent cancer was NSCLC (45%), followed by melanoma and RCC.

Overall, 93 patients (63.7%) received nivolumab in monotherapy, 42 received pembrolizumab (28.8%) and lastly 11 patients (7.5%) received atezolizumab. Sixty-three patients (43.2%) received ICI as a first line of therapy. Overall, nine patients achieved complete response (CR) (6.2%), 19 partial response (PR) (13.04%), 43 stable disease (SD) (29.5%) and the remaining 75 patients experienced progressive disease (PD) (51%).

A total of 41 patients (28.1%) were treated with steroids for an irAE during treatment with ICIs. None of the patients was treated with baseline steroid >10 mg/day prednisone or an equivalent for palliative reasons.

### 2.2. Profile of Steroids Treatment

As mentioned above, roughly 30% of patients were treated with steroids due to an irAE occurring during treatment with ICIs (data are reported in Table 2). Baseline clinical features between patients with or without irAEs treated with steroids were not significantly different. 

In total, patients who required steroid therapy experienced irAEs from six different classes: eight (19.5%) developed pneumonia, nine (22.0%) developed colitis, nine (22.0%) developed skin reactions, nine (22.0%) developed endocrine-related events, five (12.2%) developed rheumatologic events, and one patient (2.4%) developed hepatitis.

The steroid dose was 0.5–1 mg/kg/day prednisone (or equivalent methylprednisolone) in 25 patients (61.0%) and >1 mg/kg/day for 16 patients (39.0%). Prednisone was the most commonly used molecule (80.5%), followed by methylprednisolone (19.5%). The duration of steroid treatment was <14 days for 21 patients (51.2%) and >14 days for 20 patients (48.8%).

The cumulative dose of steroids (mg/kg for the days of treatment) was less than 500 mg in 25 patients (61.0%) and more than 500 mg in 16 patients (39.0%).

According to the CTCAE grades v. 4.0, registered irAEs were mainly grade 2 (we do not include irAEs grading 1 since these toxicities usually do not need steroid treatment) and have been included in a “non-serious AE” subgroup (75.6%). Ten patients (24.0%) developed a “serious” irAE (grade 3 or grade 4). Five patients (12.1%) discontinued ICIs treatment due to toxicity. We did not register any deaths due to toxicity.

We registered nine irAEs (22.0%) that occurred within the first 30 days of immunotherapy (we called this group “early steroid treatment”), while the irAEs that occurred after 30 days of therapy (“late steroid treatment”) were 32 (778.0%). In Table 3, we describe the type of irAEs divided by the onset time. 

### 2.3. Relationship between Steroids Treatment and Patient Outcome

PFS did not show any significant difference between the group of patients treated with steroids and the one without steroid therapy (*p* = 0.161). Moreover, no differences were found when patients were analyzed when comparing the administered steroid dose (more than 1 mg/kg/day, *p* = 0.166), the duration of steroid treatment (more than 14 continuous days, *p* = 0578) or the cumulative dose (more than 500 mg, *p* = 0.578).

Treatment with steroids due to an irAE within the first 30 days from the beginning of immunotherapy was not related to a shorter PFS (*p* = 0.358), while the late steroid group had significantly longer PFS (*p* = 0.045). The median PFS of the patients treated with steroids during the first 30 days was 152 days, while the median PFS of the patients not treated with steroids was 194 days (Figure 1). One of these “early steroid” patients discontinued immunotherapy due to toxicity.

Interestingly, patients who were treated with steroids for the occurrence of an irAE after 30 days from the start of immunotherapy (defined as “late steroid treatment”) had significantly longer PFS (*p* = 0.045). The median PFS of the patients treated with steroids after the first 30 days was 304 days (IQR 214—not reached) (Figure 1).

## 3. Discussion

The unique immune-activating mechanism of action of ICIs can be responsible for toxicities that result from the loss of self-tolerance and can therefore generate a plethora of auto-inflammatory events potentially involving any organ [20,21,22]. IrAEs can result from the exacerbation of a pre-existing autoimmune condition or from the induction of a new inflammatory syndrome [21,22].

IrAEs most frequently affect the skin, the gastrointestinal tract, the endocrine glands, the lungs and the liver. They rarely affect the nervous system, kidney, blood, muscles, joints, heart or eyes [21,23].

The therapeutic strategy depends on the irAEs’ severity grade, defined according to CTCAE 4.0. Usually, grade 1 events, and sometimes at a physician’s judgement even grade 2, do not require specific therapies but only symptomatic treatment [16,20]. On the other hand, the management of grade 3 or 4 AE requires moderate or high-dose systemic glucocorticoids (typically oral prednisone 1 mg/kg or equivalent or parenteral formulations) [16,21]. The steroid dose may depend on the affected organs. For example, arthralgia induced by ICIs is typically managed with lower doses (0.2–0.4 mg/Kg/day) compared with colitis or pneumonitis, which often require the administration of higher doses (0.7–1.0 mg/Kg/day) [21,24]. To maintain an anti-inflammatory effect and avoid irAE relapse, the dose of steroids should be given daily (preferably in the morning) until irAE resolution and must be then tapered gradually. Generally, a full-dose steroid treatment is usually given for 2–3 weeks, then decreased over 4–6 weeks and lastly withdrawn [21,24].

It is well known that steroid use might impair the activity of ICIs, due to its recognized immunosuppressive activity [20]. Steroids’ immune-suppressive mechanisms may act through the inhibition of the production of inflammatory mediators by immune cells including cytokines (interleukin 1 (IL-1), IL-6, Tumor Necrosis Factor (TNF) and prostaglandins (PGE-2) [25,26]. Steroids induce the resolution of inflammation, with an increase in the secretion of anti-inflammatory factors (IL-10 and Tumor Growth Factor (TGF)-β) by M2 macrophages and the increased phagocytosis of apoptotic cells. They also have effects on the adaptive immune system, suppressing CD4+T cell activation by modulating dendritic cell function and promoting the polarization of T helper (Th) cells, with the preferential differentiation of Th2 and T regulatory (Treg) cells and the inhibition of Th1 and Th17 cells. Moreover, preclinical experiences have shown that the administration of dexamethasone, whether given alone or together with anti-PD-1 therapy, leads to a considerable reduction of circulating CD4+ and CD8+ T cells [27]. In the same preclinical models, anti-PD-1 monotherapy resulted in significantly longer tumor doubling times, thus reducing the tumor volume compared to the control group treated with steroids [28]. More specifically, dexamethasone alone and the anti-PD-1 + dexamethasone combination treatment group exhibited similar results on tumor growth. Furthermore, steroids enhance the expression of PD-1 on T-cells, thereby impairing the function of activated T lymphocytes [28]. Finally, steroids induce apoptosis in hematological cells, giving strong support to their use to treat leukemias, lymphomas and myeloma [28,29]. On the contrary and for the same reasons, steroids at the beginning of immunotherapy, inhibiting the immune cascade, might impair the activation of an efficacious antitumor immune response [28,30].

In our analysis, we did not find any difference in PFS in patients treated with steroids compared to the non-steroid group. This was even true for patients who received high doses of steroids (more than 1 mg/kg). The literature about the impact of steroid administration while on ICIs is controversial. Two retrospective studies, conducted among patients affected by NSCLC, reported that baseline steroids administration was associated with a lower Objective Response Rate (ORR) and a worse PFS and OS with anti-PD-1/PD-L1 treatment [9,21,31].

However, some retrospective analyses have reported encouraging data about the use of steroids and the maintenance of the effectiveness of ICIs therapy. A retrospective study of 650 patients with NSLC treated with immunotherapy has shown that negative effects on efficacy outcomes were only seen for cancer-related indications and not for non-cancer-related indications [21,32]. More specifically, mPFS and mOS were only significantly shorter among patients who received ≥10 mg prednisone for palliative indications compared with patients who received ≥ 10 mg for cancer-unrelated reasons and with patients receiving 0 to <10 mg of (mPFS, 1.4 vs. 4.6 vs. 3.4 months, respectively; log-rank *p* < 0.001 across the three groups; mOS, 2.2 vs. 10.7 vs. 11.2 months, respectively; log-rank *p* < 0.001 across the three groups). There was no significant difference in mPFS or mOS in patients receiving steroids for non-palliative indications compared with patients receiving 0 to <10 mg of prednisone [21,32]. Interestingly, the median duration of steroid use was longer for cancer-unrelated indications, advising that the duration of steroid use before the initiation of ICI therapy does not impair anticancer efficacy [21,32].

Furthermore, in a retrospective study of 424 patients with advanced NSCLC treated with single ICI, 49 patients received steroids within the first eight weeks after the start of ICI therapy. In the 11 patients receiving steroids for non-palliative indications, the main cause for administrations were irAEs and exacerbations of chronic pulmonary obstructive disease (COPD) [21,33]. Patients receiving steroids for palliative indications had a lower median OS time (1.9 months) relative to those receiving steroids for other indications (3.4 months). Early steroids use for cancer-related symptoms proved to be an independent prognostic factor for OS [HR 4.53; 95% CI = 1.84–11.12; *p* < 0.0001] [21,33]. A meta-analysis including 16 studies with 4045 patients treated with ICIs confirmed that the use of steroids to manage adverse events did not impact OS, in contrast to their administration for disease-related symptoms, where both PFS and OS were impaired [16,21]. It is known that early steroid use for cancer-related symptoms has proven to be an independent prognostic factor for OS [NR 4.53; 95% CI = 1.84–11.12; *p* < 0.0001] [21,33].

The question raised by these analyses is whether the worst outcome associated with the administration of baseline steroids for palliative reasons is due to the frailty of patients or is due to the immunosuppressive function of steroids that impair the creation of a strong immune response in the patient.

In our analysis, when we compared the patients’ PFS by the onset of the steroid treatment (early vs. late), we did not find significantly shorter PFS in patients treated with steroids due to an irAE that occurred within the first 30 days from the start of immunotherapy. However, a meta-analysis including 16 studies with 4045 patients treated with ICIs confirmed that the use of steroids to manage adverse events did not impact OS, in contrast to their administration for disease-related symptoms, where both PFS and OS were impaired [16,21]. However, a previous retrospective analysis did not specifically report the outcome of a patient who received an early (first 4–8 weeks) administration of steroids to treat irAEs. Thus, to our knowledge, our data are the first to show that an early steroid administration to treat irAEs that occur early after the start of immunotherapy does not have a detrimental impact on patient prognosis.

Moreover, we reported that patients who were treated with steroids due to an irAE occurring 30 days after the beginning of immunotherapy had significantly longer PFS (*p* = 0.017). Importantly, this association was confirmed in a multivariate analysis, and to our knowledge this is the first study to indicate that the development of irAEs treated with steroids represents an independent predictor of ICIs’ efficacy. Intriguingly, a potential association with better prognosis in patients reporting irAEs has been previously described [34,35,36], thus balancing the immunosuppressive effect of steroid use. This is an important message to clinical oncologists, underlying the relevance of a rapid and appropriate steroid treatment in patients who experience irAEs precisely due to their better prognosis. The interpretation of our study results is mainly limited by its retrospective nature, the heterogeneity and the low number of the sample of patients. However, despite these limitations, our study might provide interesting and confirmatory results for previous reports, which could be helpful in the clinical management of patients treated with ICIs. A prospective study with larger and more homogeneous cohorts is required to confirm our findings.

## 4. Materials and Methods

### 4.1. Patients

We retrospectively analyzed data from 146 patients treated with ICIs at the Medical Oncology Unit, Careggi University Hospital (Firenze, Italy) from March 2016 to March 2020.

Patient inclusion criteria included age >18, a histologically confirmed diagnosis of metastatic NSCLC, melanoma or RCC and treatment with ICIs. Patients were treated with nivolumab, atezolizumab or pembrolizumab, depending on the histology subgroups and therapy line. Patients received treatment either until disease progression or excessive toxicity.

All occurring irAEs and data about their grade and their management with steroid treatment, including the type of steroid used, dosage and duration of therapy, were recorded in accurate case histories. The worst toxicity grades per subject will be tabulated for AEs and on-study laboratory measurements by using the National Cancer Institute (NCI) Common Terminology Criteria for Adverse Events (CTCAE) version 4.0. All patients had a measurable disease according to the Response Evaluation Criteria in Solid Tumors (RECIST v 1.1) [37], and the disease progression was confirmed 4 to 8 weeks later after first radiologic evidence of PD in clinically stable patients according to the immune (i)-RECIST [37]. 

All patients signed an informed consent form for the ICIs treatment reporting the possible occurrence of AEs and their registrations in clinical records. This study was conducted by adhering with the World Medical Association Declaration of Helsinki and was independently reviewed and approved by the Independent Ethics Committee (IEC) for Clinical Trials of the Tuscany Region (approval No.: 17332_oss). All patient data were managed in anonymity and de-identified prior to analysis.

### 4.2. Statistical Analyses

Estimates of PFS in the steroid and steroid naïve groups of patients or different subgroups were calculated using the Kaplan–Meier method, and statistical significance was examined via a log-rank test. A significance level of *p* < 0.05 was used for statistical analyses. The univariate analysis and the Cox proportional hazard model were employed to calculate the hazard ratios (HRs) and appropriate 95% CIs. Afterward, the independent effect of each parameter on PFS was investigated by a multivariate Cox regression model. All data were analyzed using the statistical software Jamovi (version 1.6, Sidney, Australia).

## 5. Conclusions

Even with some intrinsic limitations, our study may provide a valuable clinical message to oncologists. We produce interesting findings regarding the lack of association between exposure to steroids for cancer-unrelated indications and worse outcomes from anti-PD1/PD-L1. Our results are in line with those of some previous retrospective analyses reporting that the introduction of steroids within the first eight weeks of ICI therapy in patients with advanced NSLCL had no detrimental impact on the prognosis if the indication for steroid use was not related to cancer symptoms. Remarkably, despite our overall results being derived from only a single center study and from a small sample size of patients treated for early irAEs, they certainly reassure one about the use of steroids during immunotherapy and point out that systemic steroids administered to manage irAEs might not have a detrimental effect on the patient clinical outcome. 

We are planning to further assess the possible detrimental effect of baseline steroids on immunotherapy clinical outcomes and to prospectively validate our retrospective data about the absence of a negative association between the use of steroids for the management of irAEs and the efficacy of ICIs.

## Figures and Tables

**Figure 1 molecules-26-05789-f001:**
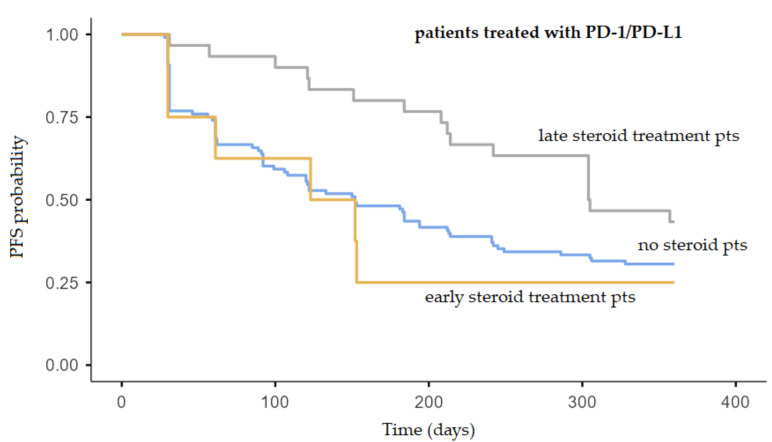
Association between steroid treatment for an irAE during the first 30 days of immunotherapy (“early treatment”), after 30 days (“late treatment”) and outcome in patients treated with anti-PD-1/PD-L1. Late steroid treatment pts vs. non-steroid pts: log-rank *p* = 0.045; early steroid treatment pts vs. non-steroid pts: log-rank *p* = 0.358). Kaplan–Meier graphs of PFS. PFS = progression-free survival; pts = patients.

**Table 1 molecules-26-05789-t001:** Clinical features of the study population.

Characteristics	No. of Patients (*n* = 146)	
Sex		
Male	98	67.1%
Female	48	32.9%
Age, years		
Average	67	
Median	70	
Range	27–91	
Tumor		
NSCLC	67	45.9%
Melanoma	46	31.5%
RCC	33	22.6%
Therapy line		
1	63	43.2%
2	70	47.9%
3	10	6.8%
4	3	2.1%
Immune checkpoint inhibitors		
Nivolumab	93	63.7%
Pembrolizumab	42	28.8%
Atezolizumab	11	7.5%
Outcome		
CR	9	6.2%
PR	19	13.0%
SD	43	29.5%
PD	75	51.4%
Steroid treatment during immunotherapy		
Yes	41	28.1%
No	105	71.9%

NSCLC = non-small cell lung cancer; RCC = renal cell cancer; CR = complete remission; PR = partial response; SD = stable disease; PD = progressive disease.

**Table 2 molecules-26-05789-t002:** Characteristic of steroids treatment and irAEs.

Characteristics of Steroids Treatment and irAEs	No. of Patients (*n* = 41)	
Cumulative dose of steroid		
<500 mg prednisone or equivalent	25	61.0%
>500 mg prednisone or equivalent	16	39.0%
Dose of steroid mg/kg		
Prednisone 0.5–1 mg/kg/day (or equivalent of methylprednisolone)	25	61.0%
Prednisone >1 mg/kg/day (or equivalent of methylprednisolone)	16	39.0%
Molecule		
Prednisone	33	80.5%
Methylprednisolone	8	19.5%
Duration of treatment		
<14 days	21	51.2%
>14 days	20	48.8%
Type of irAEs treated with steroids		
Pneumonitis	8	19.5%
Colitis	9	22.0%
Skin reactions	9	22.0%
Endocrine-related events	9	22.0%
Rheumatologic events	5	12.2%
Hepatitis	1	2.4%
irAEs grade (CTCAE v 4.0)		
Non-serious (CTCAE grade 2)	31	75.6%
Serious (CTCAE grade 3–4)	10	24.4%
Patients who discontinued ICIs due to toxicity		
ICI discontinued	5	12.1%
ICI continued	36	87.8%
Time of steroid treatment		
Early steroid treatment (first 30 days of immunotherapy)	9	22.0%
Late steroid treatment (after 30 days of immunotherapy)	32	78.0%

**Table 3 molecules-26-05789-t003:** Type of irAEs divided by onset time.

	irAEs Type						
Onset	Pulmonary	Colitis	Hepatitis	Cutaneous	Rheumatologic	Endocrine	Total
Late	4 (13.3%)	8 (26.7%)	1 (3.3%)	5 (16.7%)	3 (10.0%)	9 (30.0%)	30 (100%)
Early	4 (36.4%)	1 (9.0%)	0 (0.0%)	4 (36.4%)	2 (18.2%)	0 (0.0%)	11 (100%)
Total	8 (19.6%)	9 (21.9%)	1 (2.4%)	9 (21.9%)	5 (12.3%)	9 (21.9%)	41 (100%)

No significant difference in the occurrence of steroid treatment between different tumors (*p* = 0.192) and different ICIs (*p* = 0.671) was observed.

## Data Availability

Data available on request from the corresponding author.

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
