# Peer review of "Association of Systemic Steroid Treatment and Outcome in Patients Treated with Immune Checkpoint Inhibitors: A Real-World Analysis"

_molecules, 2021, doi:10.3390/molecules26195789_

Round 1

Reviewer 1 Report

Minor suggestions:

  1. The sentence in line 171-172 needs to be revised.
  2. Please provide a table to describe cancer-related and non-cancer-related irAE in patients receiving anti-CTLA-4 and anti-PD-1/L1, respectively.

Author Response

Reviewer 1

  1. The sentence in line 171-172 needs to be revised.

The sentence has been deleted.

  1. Please provide a table to describe cancer-related and non-cancer-related irAE in patients receiving anti-CTLA-4 and anti-PD-1/L1, respectively.

Patients with cancer experience multiple symptoms related to cancer and not to treatment. They include pain, dyspnea, fatigue, depression, and cognitive impairment and must be distinguished from adverse events, who are related to cancer treatment. In our study, we have described irAEs related to anti.PD-1/L1 treatment and in line 194-197 we have reported different types of irAEs.

Reviewer 2 Report

The authors have retrospectively evaluated the association of steroid treatment for irAEs and PFS after ICI treatment in patients with metastatic NSCLC, melanoma or RCC. The study includes data from 170 patients divided on three histologies and three overall different classes (anti-CTLA-4, anti-PD-1/-PD-L1, or the combination of anti-CTLA-4 and anti-PD-1). This makes some of the subgroups quite small and heterogenous (as the authors point out themselves) and clear conclusions are hard to make. To my mind, the most interesting conclusion is that treating irAEs with steroids more than 30 days after commencing anti-PD-1 treatment seems to be associated with a longer PFS. In addition, that early steroid treatment is not associated with a shorter PFS in anti-PD-1 treated patients.
It is indeed an area that needs further research and as such the manuscript is interesting. There are however some points that should be revised before publication can be considered.

Major points:
- Figure 2 displays PFS in patients treated with anti-CTLA-4 and from the graph it looks like there are 4 pts in the "early" group, 3 in the "late" group and assumedly 7 in the steroid naive group (as 14 patients have been indicated to have received ipilimumab alone in Table 1). However in the discussion page 8, lines 281-282 it is mentioned that the anti-CTLA-4 group is "heterogenous, including patients treated with ipilimumab in combination with anti-PD-1". Anti-CTLA-4 and the combination of anti-CTLA-4 and anti-PD-1 should not be in the same group.
- Making comparisons of PFS after anti-CTLA-4 treatment is inherently hard (assuming it is for anti-CTLA-4 treatment monotherapy, which should only include melanoma patients), as the PFS would be short in all conditions since the clinical benefit rate is expected to be lower than 25%. Despite the P-values (which are missing from the text; first paragraph page 6, lines 157-159) are not showing a significant difference, I think the graph speaks for itself regarding long-term PFS. With the low numbers the conclusions should be extremely careful.
- The majority of the discussion is more introductory in nature as there is no reference to the findings of the current study (from start of the discussion to page 8 line 254). It is not all irrelevant, but should be shortened and relevant points regarding in the context of the current study should be added.
- The conclusion should be shortened to contain only the most important conclusions.
- The manuscript would benefit from thorough proof reading. Examples in minor points.

Suggestions:
Consider comparing the accumulated dose of steroid administered if you have the data available. If you administer 1 mg/kg for 7 days it is less than 0.5 mg/kg for 21 days.

Consider including data on irAEs not requiring steroids. A considerable portion of the patients treated with steroids had grade 2 irAEs. Most likely there would be a big group with grade 2 irAEs that did NOT receive steroids. This would enable you to argue that developing irAEs requiring steroid treatment has prognostic value, but irAEs in itself does not (if the data supports it). However, I would expect that in your data grade 3/4 irAEs would be associated with a longer PFS, as the overlap with steroid treatment should be considerable.

Minor points:
Generel points: 
Corticosteroids is written in full length a couple of times before shortening to steroids. Should be noted first time and henceforth the short form should be used.
In naming the antibodies "anti" should be included with a hyphen, e.g. anti-CTLA-4.
Numbers should be separated with . instead of ,

Specific points:
Title: Immune Checkpoint Inhibitors should not be capitalized - or the rest should be for consistency.
Abstract: 
P1, LL 16-17: Non Small Cell Lung Cancer and Renal Cell Cancer are capitalized - also hyphen missing in non-small
P1, L 23: Insert "longer" in "factor for longer PFS"
Depending on how the conclusions are rephrased, this should be changed in the abstract as well.
Introduction:
P1, L34: ICIs.
P1, L36: Thus should be moved to the start of the sentence.
P2, L47: I believe "during" refers to baseline use of steroids and the sentence should be rephrased.
P2, LL67-72: Very long sentence. Consider splitting it up.
Results:
P2, L77: "in association" should be "in combination".
P3, L92: Delete "developed"
P3, L93: "Basal" should be baseline.
P3, L94: mg/die should be mg/day (this also happens later)
P4, L116: AES -> AEs
P4, L122-123: Perhaps add "(data not shown)". It is interesting that there was no difference in occurence of steroid treatment between different ICIs. I have a hard time believing the combination would not be higher had you had the numbers for reaching significance. 
P4, L133: of the patients untreated with steroids - change to not treated. Same thing multiple places, also at P5, L145.
P5, L147: Text says two groups, but three groups are described. Only data from two groups are shown though.
Methods:
P9, L312: ICI without s
P9, L313: the combination of nivo..
P9, L325: How do you collect clinical data anonymously? I think you can delete the first sentence, as the meaning is covered in the last sentence of the same paragraph.

I have not commented on the discussion and conclusion section, as I think most of it should be rewritten.

Table 2:
In the classification of irAEs it should state (CTCAE grade 2) etc
Change to "Patients who discontinued ICIs due to toxicity"

Table 3: 
I dont think it makes a lot of sense to test the endocrine irAEs early vs late. Were all conditions tested? It is well known that endocrine AEs most often develop later in therapy.

Author Response

Reviewer 2

The authors have retrospectively evaluated the association of steroid treatment for irAEs and PFS after ICI treatment in patients with metastatic NSCLC, melanoma or RCC. The study includes data from 170 patients divided on three histologies and three overall different classes (anti-CTLA-4, anti-PD-1/-PD-L1, or the combination of anti-CTLA-4 and anti-PD-1). This makes some of the subgroups quite small and heterogenous (as the authors point out themselves) and clear conclusions are hard to make. To my mind, the most interesting conclusion is that treating irAEs with steroids more than 30 days after commencing anti-PD-1 treatment seems to be associated with a longer PFS. In addition, that early steroid treatment is not associated with a shorter PFS in anti-PD-1 treated patients. It is indeed an area that needs further research and as such the manuscript is interesting. There are however some points that should be revised before publication can be considered.

Major points:

- Figure 2 displays PFS in patients treated with anti-CTLA-4 and from the graph it looks like there are 4 pts in the "early" group, 3 in the "late" group and assumedly 7 in the steroid naive group (as 14 patients have been indicated to have received ipilimumab alone in Table 1). However in the discussion page 8, lines 281-282 it is mentioned that the anti-CTLA-4 group is "heterogenous, including patients treated with ipilimumab in combination with anti-PD-1". Anti-CTLA-4 and the combination of anti-CTLA-4 and anti-PD-1 should not be in the same group.

According to the reviewer suggestion  we have removed data relative to patients treated with CTLA4 in mono or in combination due the small size of this subgroup. Accordingly we have performed new statistical analysis.

- Making comparisons of PFS after anti-CTLA-4 treatment is inherently hard (assuming it is for anti-CTLA-4 treatment monotherapy, which should only include melanoma patients), as the PFS would be short in all conditions since the clinical benefit rate is expected to be lower than 25%. Despite the P-values (which are missing from the text; first paragraph page 6, lines 157-159) are not showing a significant difference, I think the graph speaks for itself regarding long-term PFS. With the low numbers the conclusions should be extremely careful.

See above.

- The majority of the discussion is more introductory in nature as there is no reference to the findings of the current study (from start of the discussion to page 8 line 254). It is not all irrelevant, but should be shortened and relevant points regarding in the context of the current study should be added.

According to Reviewer suggestion we have extensively edited the Discussion section of the revised version of the manuscript.

- The conclusion should be shortened to contain only the most important conclusions.

According to Reviewer suggestion we have edited the Conclusion of the revised version of the manuscript.

- The manuscript would benefit from thorough proof reading. Examples in minor points. The manuscript has been extensively edited.

Suggestions:

Consider comparing the accumulated dose of steroid administered if you have the data available. If you administer 1 mg/kg for 7 days it is less than 0.5 mg/kg for 21 days.

 It has been add in the new version of the manuscript.

Consider including data on irAEs not requiring steroids. A considerable portion of the patients treated with steroids had grade 2 irAEs. Most likely there would be a big group with grade 2 irAEs that did NOT receive steroids. This would enable you to argue that developing irAEs requiring steroid treatment has prognostic value, but irAEs in itself does not (if the data supports it). However, I would expect that in your data grade 3/4 irAEs would be associated with a longer PFS, as the overlap with steroid treatment should be considerable. We are planning to analyze data on irAEs not requiring steroids and the role of other concomitant medications in future studies

Minor points:
General points: 
Corticosteroids is written in full length a couple of times before shortening to steroids. Should be noted first time and henceforth the short form should be used.

It has been revised in the new version of the manuscript according to the reviewer suggestion.

In naming the antibodies "anti" should be included with a hyphen, e.g. anti-CTLA-4.

It has been revised in the new version of the manuscript.

Numbers should be separated with . instead of ,It has been revised in the new version of the manuscript.

Specific points:
Title: Immune Checkpoint Inhibitors should not be capitalized - or the rest should be for consistency.

 It has been revised.

Abstract: 
P1, LL 16-17: Non Small Cell Lung Cancer and Renal Cell Cancer are capitalized - also hyphen missing in non-small.

It has been revised.

P1, L 23: Insert "longer" in "factor for longer PFS"

It has been removed.

Depending on how the conclusions are rephrased, this should be changed in the abstract as well.

The abstract has been edited.

Introduction:
P1, L34: ICIs.

It has been revised.

P1, L36: Thus should be moved to the start of the sentence.

 It has been revised.

P2, L47: I believe "during" refers to baseline use of steroids and the sentence should be rephrased.

The sentence has been rephrased.

P2, LL67-72: Very long sentence. Consider splitting it up.

The sentence has been splitted.

Results:
P2, L77: "in association" should be "in combination".

It has been revised.

P3, L92: Delete "developed"

It has been deleted.

P3, L93: "Basal" should be baseline.

It has been revised.

P3, L94: mg/die should be mg/day (this also happens later).

 It has been revised.

P4, L116: AES -> AEs

It has been revised.

P4, L122-123: Perhaps add "(data not shown)". It is interesting that there was no difference in occurence of steroid treatment between different ICIs. I have a hard time believing the combination would not be higher had you had the numbers for reaching significance.

Due the small size of the subgroup we have removed data relative to patients treated with CTLA-4 in mono or in combination.

P4, L133: of the patients untreated with steroids - change to not treated. Same thing multiple places, also at P5, L145.

 We have edited.

P5, L147: Text says two groups, but three groups are described. Only data from two groups are shown though. It has been revised.

Methods:
P9, L312: ICI without s.

It has been revised.

P9, L313: the combination of nivo.

It has been revised.

P9, L325: How do you collect clinical data anonymously? I think you can delete the first sentence, as the meaning is covered in the last sentence of the same paragraph.

It has been deleted.

I have not commented on the discussion and conclusion section, as I think most of it should be rewritten.

According to the reviewer discussion and conclusion section have been edited.

Table 2:
In the classification of irAEs it should state (CTCAE grade 2) etc
Change to "Patients who discontinued ICIs due to toxicity"

It has been changed.

Table 3: 
I dont think it makes a lot of sense to test the endocrine irAEs early vs late. Were all conditions tested? It is well known that endocrine AEs most often develop later in therapy.

It has been removed.

Reviewer 3 Report

The manuscript by Paderi et al. entitled “Association of systemic steroid treatment and outcome in patients treated with immune checkpoint inhibitors: a real-world analysis” identifies a relevant area for investigation. The authors highlight the need for additional studies as the results reported in the literature are contradictory in this regard. Hence, the manuscript addresses an important issue (immune-related adverse events) with respect to cancer patients who are undergoing immunotherapy. However, the manuscript, in its current form, has several problems. The manuscript needs to be revised on multiple fronts to clarify and qualify the study’s impact and meaning.

Abstract: the abstract needs to reflect the data presented in the results section.

Introduction: the rationale of the study is not clear. For example, lines 55-72: too wordy.

Line 49: inhibitors treatment?

Line 70: RRC?

The manuscript needs editing as the current version is filled with run-on sentences, format problems and grammatical mistakes.

Line 160: did not shown?

Line 179: hereby?

Line 183: sentence format?

Lines 190-191: Rarely, they interest the nervous system?

Line 192: defined according CTCAE?

Line 205: due its?

Line 219: More in details?

Line 233: In detail?

There is no flow in the Discussion section and should also be trimmed.

Lines 183-253: ?

Line 277: word?

Lines 276-280: long sentence.

Line 283: More in detail?

Lines 283-287: sentence format?

Lines 294-297: sentence format?

Basal steroid?

Author Response

Reviewer 3

The manuscript by Paderi et al. entitled “Association of systemic steroid treatment and outcome in patients treated with immune checkpoint inhibitors: a real-world analysis” identifies a relevant area for investigation. The authors highlight the need for additional studies as the results reported in the literature are contradictory in this regard. Hence, the manuscript addresses an important issue (immune-related adverse events) with respect to cancer patients who are undergoing immunotherapy. However, the manuscript, in its current form, has several problems. The manuscript needs to be revised on multiple fronts to clarify and qualify the study’s impact and meaning.

Abstract: the abstract needs to reflect the data presented in the results section.

Introduction: the rationale of the study is not clear. For example, lines 55-72: too wordy.

It has been revised

Line 49: inhibitors treatment?  It has been revised Line 70: RRC?

RRC has been replaced with RCC.

The manuscript needs editing as the current version is filled with run-on sentences, format problems and grammatical mistakes.

Line 160: did not shown?

It has been revised

Line 179: hereby?

It has been revised

Line 183: sentence format?

It has been removed.

Lines 190-191: Rarely, they interest the nervous system?

It has been revised

Line 192: defined according CTCAE?

 It has been revised

Line 205: due its?

 It has been revised

Line 219: More in details?

 It has been revised

Line 233: In detail?

It has been revised

There is no flow in the Discussion section and should also be trimmed.

The Discussion section has been revised.

Lines 183-253. Is has been revised.Line 277: word?

 It has been removed

Lines 276-280: long sentence.

 It has been removed

Line 283: More in detail?  

It has been removed.

Lines 283-287: sentence format?

It has been deleted.

Lines 294-297: sentence format?

It has been deleted.

Basal steroid?

It has been changed with baseline

Round 2

Reviewer 2 Report

Abstract :
L 25: End sentence with period.

Introduction:
L34: Remove second "thus"
L46: Remove space between anti- and programmed death ligand-1
L63: "In light of these evidences" is not correct. Either it should be "this evidence" or perhaps preferable "these observations"

Discussion:
L150: Rarely, the affect - instead of interest
L185: One period to many at the end of sentence
L202-204: Sentence starting with "There was no significant.." is weird. Rewrite.
L222: String should be strong.
L246: Smallness of the sample of should be changed to low number of

Table 2:
Prednisone 0.5-1 group missing a space between of and methylprednisolone
I would change the order so the text in the hyphen is "CTCAE grade 2" and ".. 3-4"
Consider writing just 22% - in the corresponding text 22.04% is indicated. Use either or

Table 3:
Second part the total is n = 49. In the text and first part of the table it is n = 41.

Overall:
Cumulative dose of prednisolone is now included in Table 2, however no statistical testing of whether this impacts clinical outcome has been included. It would strengthen the manuscript to include such an analysis.

Author Response

We have revised accordingly to reviewer suggestions.

Abstract :
L 25: End sentence with period. It has been edited.

Introduction:
L34: Remove second "thus". It has been removed.
L46: Remove space between anti- and programmed death ligand-1. It has been removed.
L63: "In light of these evidences" is not correct. Either it should be "this evidence" or perhaps preferable "these observations". It has been edited.

Discussion:
L150: Rarely, the affect - instead of interest. It has been edited.
L185: One period to many at the end of sentence. It has been edited.
L202-204: Sentence starting with "There was no significant.." is weird. Rewrite. It has been edited.
L222: String should be strong. It has been edited.
L246: Smallness of the sample of should be changed to low number of. It has been edited.

Table 2:
Prednisone 0.5-1 group missing a space between of and methylprednisolone
I would change the order so the text in the hyphen is "CTCAE grade 2" and ".. 3-4"
Consider writing just 22% - in the corresponding text 22.04% is indicated. Use either or

It has been edited.

Table 3:
Second part the total is n = 49. In the text and first part of the table it is n = 41.

It has been edited.

Overall:
Cumulative dose of prednisolone is now included in Table 2, however no statistical testing of whether this impacts clinical outcome has been included. It would strengthen the manuscript to include such an analysis. It has been edited.